# Negative regulation of type I interferon signaling by integrin-linked kinase permits dengue virus replication

Yi-Sheng Kao[1]☯, Li-Chiu Wang[2]☯, Po-Chun Chang[1], Heng-Ming Lin[1], Yee-Shin Lin[1,3,4], Chia-Yi Yu[4,5], Chien-Chin Chen[6,7], Chiou-Feng Lin[8], Trai-Ming Yeh[4,9], Shu-Wen Wan[1,3,4], Jen-Ren Wang[4,9], Tzong-Shiann Ho[4,10], Chien-Chou Chu[1], Bo-Cheng Zhang[1], Chih-Peng Chang[1,3,4] *

1 Department of Microbiology & Immunology, College of Medicine, National Cheng Kung University, Tainan, Taiwan, 2 School of Medicine, I-Shou University, Kaohsiung, Taiwan, 3 The Institute of Basic Medical Sciences, College of Medicine, National Cheng Kung University, Tainan, Taiwan, 4 Center of Infectious Disease and Signaling Research, National Cheng Kung University, Tainan, Taiwan, 5 National Institute of Infectious Diseases and Vaccinology, National Health Research Institutes, Tainan, Taiwan, 6 Department of Pathology, Ditmanson Medical Foundation Chia-Yi Christian Hospital, Chiayi, Taiwan, 7 Department of Cosmetic Science, Chia Nan University of Pharmacy and Science, Tainan, Taiwan, 8 Graduate Institute of Medical Sciences, College of Medicine, Taipei Medical University, Taipei, Taiwan, 9 Department of Medical Laboratory Science and Biotechnology, College of Medicine, National Cheng Kung University, Tainan, Taiwan, 10 Department of Pediatrics, National Cheng Kung University Hospital, College of Medicine, National Cheng Kung University, Tainan, Taiwan

☯ These authors contributed equally to this work.
* cpchang@mail.ncku.edu.tw

**Data Availability Statement:** All relevant data are within the manuscript and its Supporting Information files.

## Abstract

Dengue virus (DENV) infection can induce life-threatening dengue hemorrhagic fever/dengue shock syndrome in infected patients. DENV is a threat to global health due to its growing numbers and incidence of infection in the last 50 years. During infection, DENV expresses ten structural and nonstructural proteins modulating cell responses to benefit viral replication. However, the lack of knowledge regarding the cellular proteins and their functions in enhancing DENV pathogenesis impedes the development of antiviral drugs and therapies against fatal DENV infection. Here, we identified that integrin-linked kinase (ILK) is a novel enhancing factor for DENV infection by suppressing type I interferon (IFN) responses. Mechanistically, ILK binds DENV NS1 and NS3, activates Akt and Erk, and induces NF-κB-driven suppressor of cytokine signaling 3 (SOCS3) expression. Elevated SOCS3 in DENV-infected cells inhibits phosphorylation of STAT1/2 and expression of interferon-stimulated genes (ISGs). Inhibiting ILK, Akt, or Erk activation abrogates SOCS3 expression. In DENV-infected mice, the treatment of an ILK inhibitor significantly reduces viral loads in the brains, disease severity, and mortality rate. Collectively, our results show that ILK is a potential therapeutic target against DENV infection.

## Author summary

Dengue virus (DENV) can alter cell responses to benefit viral replication. However, most cellular proteins, especially those enhancing viral replication, remains unknown, and their

**Funding:** This work was supported by MOST Grants MOST 104-2320-B-006 -038 -MY3; 109-2327-B-006-010; 110-2327-B-006-005 (CPC) and NHRI Grant NHRI-109A1-MRCO-02202013 (CPC). The funders had no role in study design, data collection and analysis, decision to publish, or preparation of the manuscript.

**Competing interests:** The authors have declared that no competing interests exist.

roles in DENV pathogenesis are elusive. Here, we show that integrin-linked kinase (ILK) enhanced DENV infection. ILK binds DENV NS1 and NS3 to induce SOCS3 expression and abrogate STAT1/2-mediated expression of interferon-stimulated genes via the Akt-Erk- NF-κB pathway. Knockdown or inhibiting ILK enhances SOCS3 expression and reduces DENV yields in cells. Furthermore, inhibiting ILK in DENV-infected mice significantly decreased viral loads in the mouse brain and mortality. In conclusion, we identified ILK as a potential therapeutic target for DENV infection.

## Introduction

Dengue virus (DENV) in the *Flaviviridae* family is the most common cause of mosquito-borne diseases transmitted by *Aedes aegypti* and *Aedes albopictus* mosquitoes. DENV infection is often asymptomatic or induces a wide range of self-limited symptoms, including fever, skin rashes, and pains behind the eyes. Occasionally, patients can develop fatal dengue hemorrhagic fever/dengue shock syndrome (DHF/DSS). It is believed that the overt production in patients of cytokines and chemokines, such as interleukin (IL)-1β, IL-4, IL-6, IL-8, tumor necrosis factor (TNF)-α, CXCL10, and macrophage migration inhibitory factor (MIF), leads to DHF/DSS [1–4]. According to a report of the World Health Organization, the incidence of dengue has increased by 30-fold over the last 50 years. Currently, it is estimated that there are 390 million dengue cases per year [5]. More than 100 countries worldwide include endemic regions of dengue, including Taiwan. The recent outbreaks in Taiwan from 2014 to 2015 caused 60,000 cases of dengue fever and 250 deaths [6]. Despite the fast-emerging and wide-spread nature of dengue infection, there are as yet no antiviral drugs or therapies available to prevent or ameliorate DENV-induced diseases.

There are four serotypes of DENV in which DENV2 infection is more likely to induce severe diseases [7]. Mononuclear phagocytes are the primary target cells for DENV infection in humans [8], and several cell lines support DENV infection *in vitro*, such as A549, Huh-7, and U937 cells. During infection, DENV binds to cellular receptors, including β1- and β3-integrin [9,10] and DC-SIGN [11], to facilitate virus binding and entry. After entry, its positive, single-stranded RNA is translated into a polypeptide and processed by cellular and viral proteases to yield three structural (E, envelope; prM, precursor membrane; C, capsid) and seven nonstructural (NS) proteins (NS1, NS2A, NS2B, NS3, NS4A, NS4B, and NS5). DENV infection activates Akt by phosphorylating both Thr308 and Ser473 to enhance virus replication [12–14]. Furthermore, the NS proteins interact with cellular proteins to help DENV infection by altering ER and autophagic membrane to form replication complexes or by suppressing antiviral responses [15–17]. However, most of the cellular proteins bound to NS proteins and their roles in DENV pathogenesis remain elusive.

DENV infection elicits innate immune recognition by Toll-like receptor (TLR) 3, 7, and 8 in endosomes and retinoic acid-inducible gene I (RIG-I) and melanoma differentiation-associated gene 5 (MDA-5) in the cytoplasm [18–20]. Recognition of DENV by these receptors activates nuclear factor κ B (NF-κB) and interferon (IFN) regulator factors 3 or 7 to produce proinflammatory cytokines and type I (α and β) and III (λ) IFN, respectively. Secreted IFN binds to receptors and induces phosphorylation of Janus kinase (JAK) and signal transducer and activator of transcription (STAT) 1 and 2. The phosphorylated STAT1/2 translocates into the nucleus and induces IFN-stimulated gene (ISG) expression to restrict virus infection. The suppressor of cytokine signaling (SOCS) proteins, especially SOCS1 and SOCS3, are endogenous negative regulators that suppress JAK-mediated STAT1/2 phosphorylation [21]. DENV

is shown to increase SOCS1 and SOCS3 expression in infected epithelial cells [22] and mononuclear phagocytes [23] to suppress antiviral responses. In patients with DHF, the level of SOCS3 in the blood is significantly elevated and is suggested to be a predictive biomarker [24]. However, few studies focus on how DENV increases SOCS expression.

Integrin-linked kinase (ILK) is a 59-kDa serine/threonine protein kinase binding on the cytoplasmic domain of β1 integrin [25]. ILK comprises three conserved domains, including N-terminal ankyrin (ANK) repeats, a central pleckstrin homology (PH)-like domain, and a C-terminal kinase domain [26]. ILK functions as an adaptor protein to link integrins with cytoskeletons. As a kinase, ILK is activated by phosphoinositide 3-kinase (PI3K) to phosphorylate Akt on Ser473, GSK3β on Ser9, or NF-κB subunit p65 on Ser539 [27–29]. The phosphorylated p65 is re-localized into the nucleus to initiate NF-κB-regulated gene expression. Accordingly, ILK plays a vital role in regulating various cellular processes, even virus infection. Currently, although it has been revealed that ILK participates in RNA and DNA virus infection [30–32], its role in DENV infection has not yet been addressed.

In this study we have identified that ILK enhances DENV infection. ILK binds DENV NS1 and NS3 to enhance SOCS3 expression via the Akt-Erk- NF-κB pathway. The enhanced SOCS3 expression results in decreased STAT1/2-driven ISG expression and increased DENV replication. Blocking ILK by the inhibitor, OSU-T315, significantly improves the survival rate of DENV-infected mice, with reduced viral loads in the mouse brains and reduced disease severity.

## Results

### ILK knockdown reduces DENV infection

Previous studies showed that DENV infection activates Akt, and Akt inhibitors reduce DENV infection [12–14]. We determined Akt phosphorylation (p-) on Ser473 (p-Akt) in DENV serotype 2 (DENV2)-infected A549 cells and found an increase in phosphorylated Akt (**S1A Fig**). While it is known that ILK phosphorylates Akt on Ser473 [27,28], the roles of ILK in DENV infection remain elusive. To address this issue, we established A549 cells with control shRNA (shLuc) or shRNA targeting ILK 3'UTR (shILK). The ILK-specific shRNA abolished corresponding protein expression (**Fig 1A**) without affecting cell viability (**S1B Fig**). In infected control cells, the p-Akt levels were increased from 3 to 24 hours post-infection (hpi), and viral proteins NS1 and NS4B were detected at 24 hpi (**Fig 1A**). ILK knockdown attenuated p-Akt levels about 1.5-fold in mock-infected cells and infected cells from 3 to 24 hpi as well as NS1 and NS4B in infected cells at 24 hpi. OSU-T315 is an ILK inhibitor suppressing ILK-mediated Akt phosphorylation on Ser473 [33]. Treatment with 1.5 to 3 μM OSU-T315 in A549 cells diminished p-Akt level without affecting cell viability (**S1C** and **S1D Fig**, respectively). In infected A549 cells, OSU-T315 decreased the expression of viral proteins, including NS1, NS3, NS4B, and prM, in a dose-dependent manner (**Fig 1B**). These results show that knockdown or inhibition of ILK reduced Akt phosphorylation and DENV2 infection.

To determine if DENV2 yields were reduced in ILK knockdown cells, we detected double-stranded RNA (dsRNA), an intermediate generated during DENV genome duplication, in infected cells using immunofluorescence staining. ILK knockdown not only diminished the dsRNA intensity in cells but also reduced the percentage of dsRNA-positive cells by 50% (**Fig 1C**). The mean viral yield was lower in ILK knockdown cells than in control cells by 55-fold. Similar results were observed in cells infected with other serotypes of DENV (**Fig 1D**), showing that ILK knockdown reduces DENV infection independent of virus serotypes.

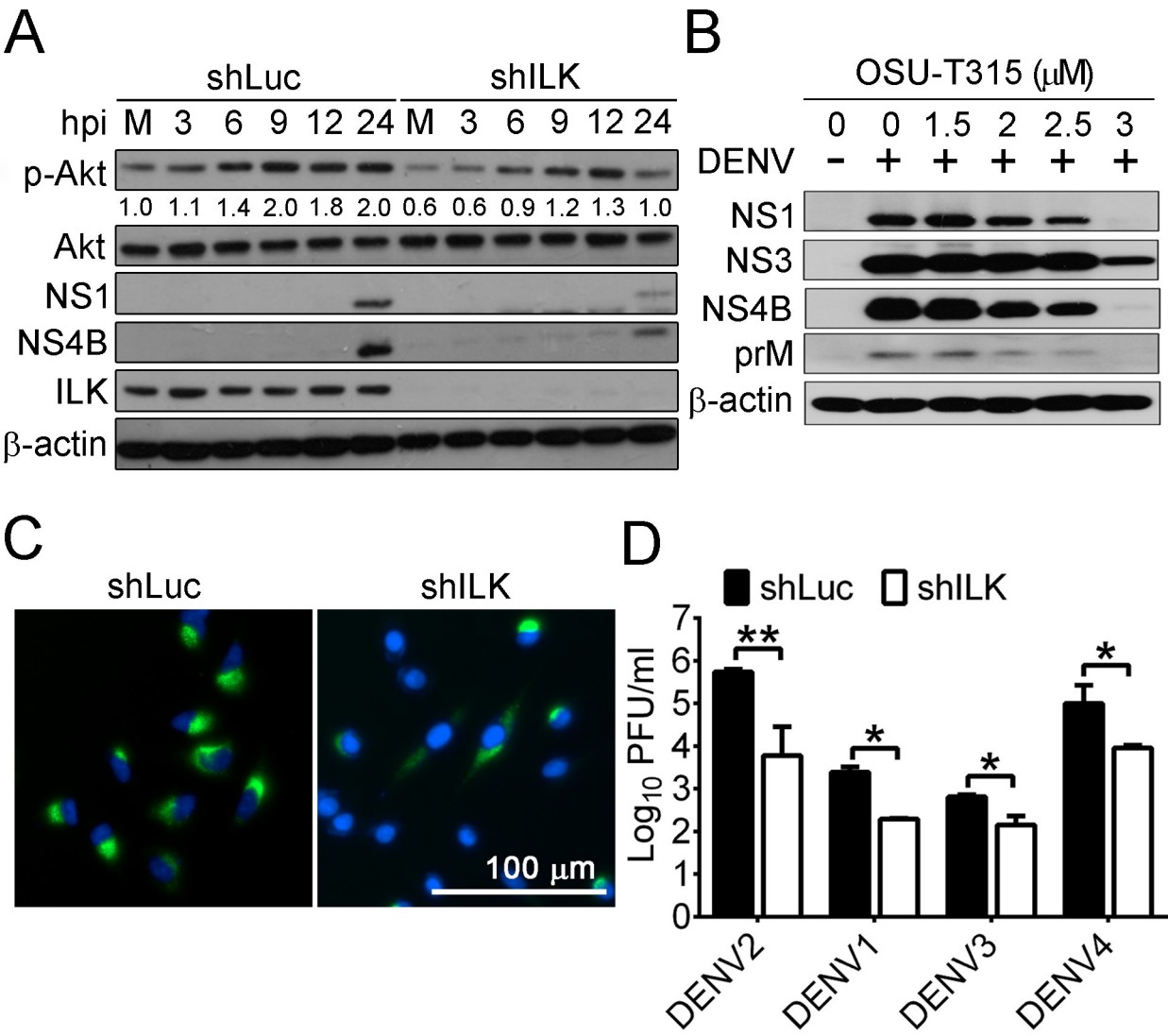

**Fig 1. ILK knockdown reduces DENV infection.** (**A**) Representative western blots of phosphorylated (p-) and indicated proteins in the mock- (M) and DENV serotype 2 (DENV2)-infected control (shLuc) and ILK knockdown (shILK) A549 cells at indicated hours post-infection (hpi) are shown. The ratios of phosphorylated to total Akt relative to mock-infected control cells are shown. (**B**) Representative western blots of indicated proteins in A549 cells infected without (-) or with (+) DENV2 and treated with indicated concentrations of OSU-T315 at 24 hpi are shown. (**C**) Representative images of DENV2-infected A549 cells stained with anti-dsRNA antibody (green) at 24 hpi are shown. Nuclei were counterstained with DAPI. (**D**) The viral yields in indicated control and ILK knockdown A549 cells infected with indicated serotypes of DENV at 24 hpi are shown. Data represent means + SD (error bars) in panel D. $^*p < 0.05$, $^{**}p < 0.01$.

## ILK knockdown reduces DENV infection after virus binding and entry

To identify DENV infection steps controlled by ILK, we first assessed the levels of DENV binding to control and ILK knockdown cells. Briefly, A549 cells were infected with DENV at a multiplicity of infection (MOI) of 25 for 2 hours at 4°C, a temperature that allows virus binding but not entry, washed with PBS to remove the unbound virus, stained with an antibody against viral envelope (E) protein, and analyzed by flow cytometry as previously described [12]. The fluorescence intensity of cells stained with anti-E antibody (**S2A Fig**) and the percentage of virus-bound (E-positive) cells (**S2B Fig**) were comparable in control and ILK knockdown cells, indicating that ILK does not play a role in virus binding.

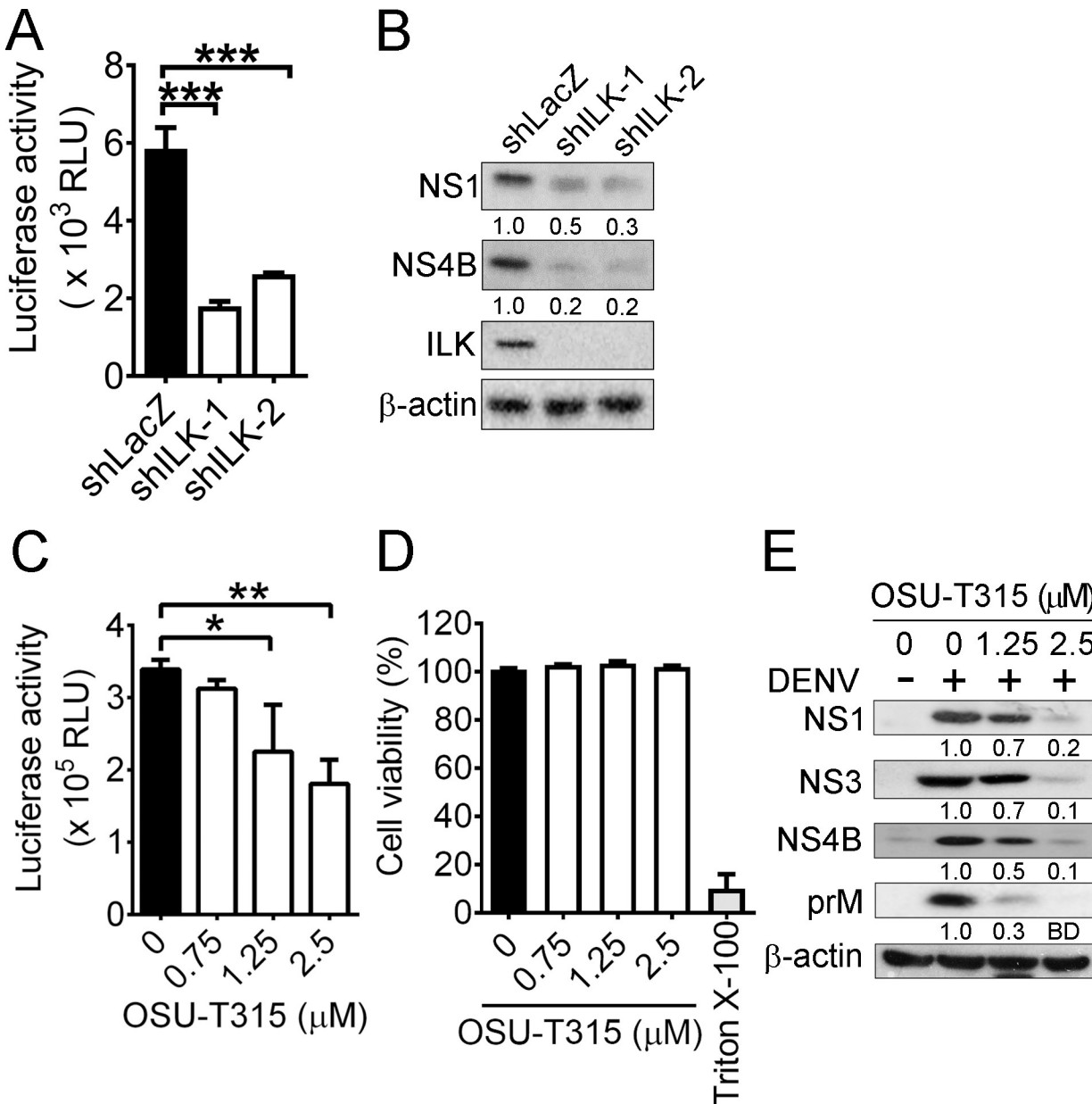

**Fig 2. ILK knockdown reduces DENV infection after virus binding and entry.** (**A** and **B**) The luciferase activity (**A**) and representative western blots of indicated proteins (**B**) in Huh7 cells with DENV replicon expressing luciferase and control shRNA (shLacZ) or shRNA targeting ILK 3'UTR (shILK-1) or CDS (shILK-2) are shown. (**C** and **D**) The luciferase activity (**C**) and cell viability (**D**) of Huh7 replicon cells treated with indicated concentrations of OSU-T315 or 0.05% Triton X-100 for 24 hours are shown. The viability of cells without OSU-T315 treatment was set as 100%. (**E**) Representative western blots of indicated proteins in Huh7 cells infected without (-) or with (+) DENV and treated with the indicated concentration of OSU-T315 for 24 hours are shown. The ratios of viral protein to β-actin relative to infected control (shLacZ) cells or infected cells without OSU-T315 treatment are shown. Data represent mean + SD (error bars) in bar graphs. $^*p < 0.05$, $^{**}p < 0.01$, $^{***}p < 0.001$. RLU, relative light unit. BD, below detection.

Next, we assessed the effect of ILK in DENV infection using a previously established Huh7 cell line carrying DENV replicon expressing firefly luciferase [34], in which virus entry is bypassed. The Huh7 replicon cells were transduced with control shRNA (shLacZ) or shRNA targeting ILK 3'UTR (ILK-1) or CDS (ILK-2). In Huh7 replicon cells with knocked-down ILK, the luciferase activity (**Fig 2A**) and NS1 and NS4B expressed by DENV replicon (**Fig 2B**) were

dramatically decreased when compared with those in control cells. Treatment with OSU-T315 inhibited luciferase activity with significant differences found at 1.25 and 2.5 μM (**Fig 2C**) without affecting cell viability (**Fig 2D**). Consistently, OSU-T315 with a concentration of 1.25 and 2.5 μM attenuated the expression of viral proteins in DENV-infected Huh7 cells (**Fig 2E**). Accordingly, these findings show that ILK knockdown decreases DENV replication after virus entry.

DENV has been known to induce autophagy to benefit viral replication [35–37]. The PI3K inhibitors decrease autophagosome formation and DENV replication [35,38]. Since ILK is activated by PI3K, we determined whether ILK facilitates autophagosome formation in DENV-infected cells by detecting LC3 I/II conversion. After 3 hpi, DENV infection increased the conversion of LC3 I to II in control cells. However, there was no difference in the LC3 I/II conversion between infected control and ILK knockdown cells (**S3A Fig**), indicating that ILK facilitates DENV replication in a manner independent of autophagosome formation. Additionally, activation of Akt has been reported to facilitate DENV replication by blocking apoptosis [13]. Thus, we determined the levels of cleaved caspase-3, a marker of apoptotic cell death, in DENV-infected control or ILK knockdown cells. In control cells, the cleaved caspase-3 was detected concomitantly with NS4B expression at 24 hpi. Nonetheless, there was no increase in the cleaved caspase-3 levels in ILK knockdown cells after infection (**S3B Fig**).

## ILK regulates SOCS3 expression to attenuate IFN signaling

Type I and III IFNs stimulate the expression of ISGs and suppress virus replication. Therefore, we assayed if ILK suppresses the induction of type I (α and β) and III (λ) IFNs and ISGs (MxA, ISG54, and ISG56) in DENV-infected cells. At 24 hpi, the levels of IFN-α, -β, and -λ in control cells were significantly increased and were not different from those in ILK knockdown cells (**Figs 3A** and **S4A** and **S4B**). However, the mRNA level of MxA was significantly increased in ILK knockdown cells before and, especially, after 24 hours of DENV infection by 2- and 60-fold, respectively (**Fig 3B**). Similar results were observed in the mRNA levels of ISG54 and ISG56 (**S4C and S4D Fig**), showing that ILK knockdown enhances ISG rather than type I/III IFN expression. We further assayed the levels of phosphorylated (p-) STAT1 and STAT2 in control and ILK knockdown cells. Upon IFN-β stimulation, p-STAT1 and p-STAT2 were increased in control and ILK knockdown cells, with higher levels found in ILK knockdown cells (**S5 Fig**). In DENV-infected cells, p-STAT1 and p-STAT2 were detected at 24 hpi, with levels further enhanced by ILK knockdown (**Fig 3C**). Treatment of OSU-T315 enhanced STAT1 phosphorylation and MxA expression and reduced NS1 expression in infected cells (**Fig 3D**). These results show that ILK attenuates STAT1/2-activated ISG expression.

Previous studies showed that SOCS1 and SOCS3 suppress IFN signaling in DENV-infected cells [22,23]. We found that DENV infection increased SOCS3 mRNA and protein levels, while SOCS1 expression was unaffected (Figs **S4E** and **4A**, respectively). SOCS3 knockdown via shRNA abolished SOCS3 mRNA expression (**Fig 4B** upper panel), dramatically increased ISG54 and ISG56 mRNA levels and STAT1 phosphorylation in cells without DENV infection (**Fig 4B** lower panel and **Fig 4C**, respectively), and reduced NS1 and NS4B expression and viral yields in infected cells (**Fig 4C and 4D**). These results show that DENV enhances SOCS3 expression to attenuate IFN signaling. Infection of the UV-inactivated DENV, a virus able to bind and enter cells but unable to replicate, failed to increase SOCS3 expression in cells (**S6 Fig**), indicating the essential role of DENV replication in SOCS3 induction.

We further assessed if ILK regulates SOCS1 and SOCS3 expression. In mock-infected cells, ILK knockdown abrogated SOCS3 but not SOCS1 expression (**Fig 4A**), indicating that ILK is required for SOCS3 expression. Notably, ILK knockdown and OSU-T315 treatment

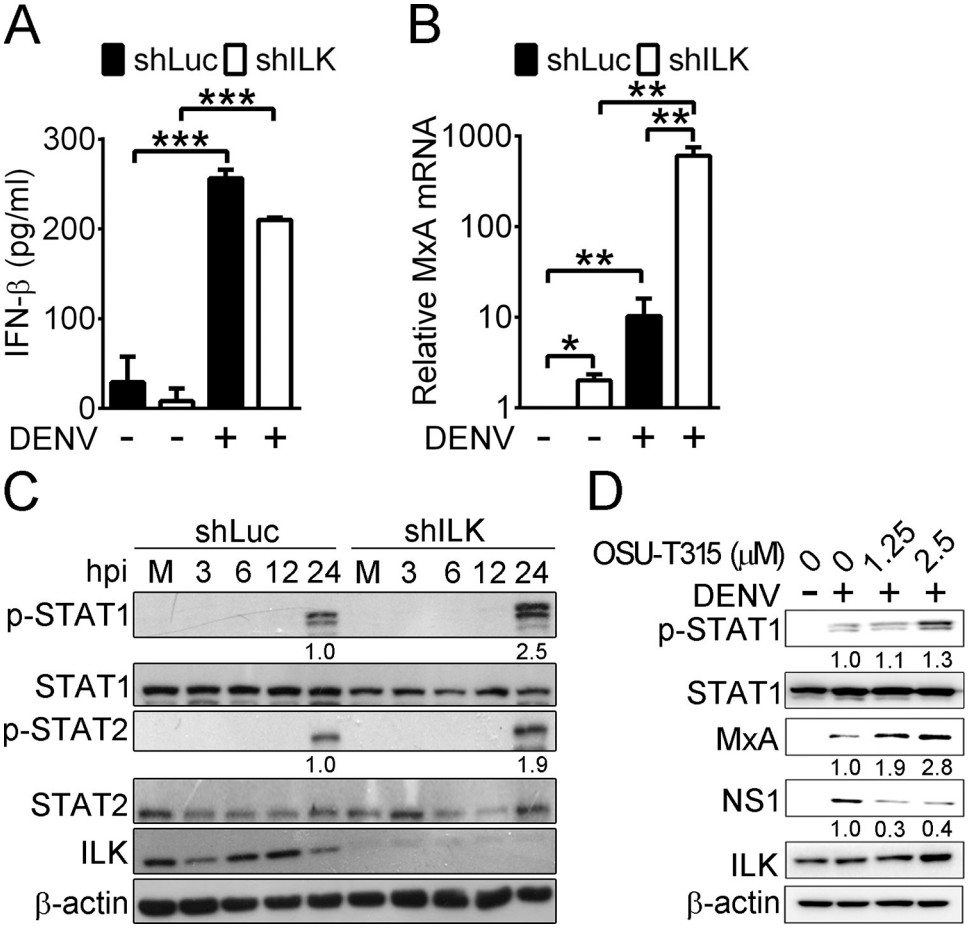

**Fig 3. ILK attenuates IFN-induced STAT1/2 activation and ISG expression.** (**A-B**) The protein levels of IFN-β (**A**) and mRNA levels of MxA (**B**) in control (shLuc) or ILK knockdown (shILK) A549 cells infected without (-) or with (+) DENV at 24 hours post-infection (hpi) are shown. The MxA level in control cells without DENV infection was set as 1. Data represent mean + SD (error bars). $^*p < 0.05$, $^{**}p < 0.01$, $^{***}p < 0.001$. (**C**) Representative western blots of phosphorylated (p-) and indicated proteins in mock- (M) and DENV-infected control or ILK knockdown A549 cells at indicated hpi are shown. The ratios of phosphorylated to total STAT1 and STAT2 relative to infected control cells at 24 hpi are shown. (**D**) Representative western blots of indicated proteins in A549 cells infected without (-) or with (+) DENV and treated with the indicated concentration of OSU-T315 for 24 hours are shown. The ratios of MxA and NS1 to β-actin as well as phosphorylated to total STAT1 relative to infected cells without OSU-T315 treatment are shown.

diminished DENV-induced SOCS3 expression (**Figs 4A and S4E** and **4E**). Similar results were observed in U937 cells, the primary target for DENV infection. Knocking down ILK in U937 cells reduced SOCS3 and viral protein expression and viral yields and enhanced STAT1 phosphorylation at 24 hpi (**S7 Fig**). The results suggest that DENV hijacks ILK-regulated SOCS3 expression to attenuate type I IFN signaling.

## ILK regulates SOCS3 expression via Akt-Erk-NF-κB pathway

Influenza A virus infection-induced SOCS3 expression is dependent on NF-κB activation [39]. Interestingly, blocking ILK reduces NF-κB activity in cells by attenuating phosphorylation of subunit p65 on Ser536 [29]. Accordingly, we aimed to determine if ILK regulates SOCS3 expression via NF-κB. Our data showed that DENV infection increased the phosphorylation of p65 on Ser536 (p-p65) in control cells at 3 to 24 hpi, while ILK knockdown abrogated p65 phosphorylation in mock- and DENV-infected cells (**Fig 5A**). Additionally, treatment of NF-

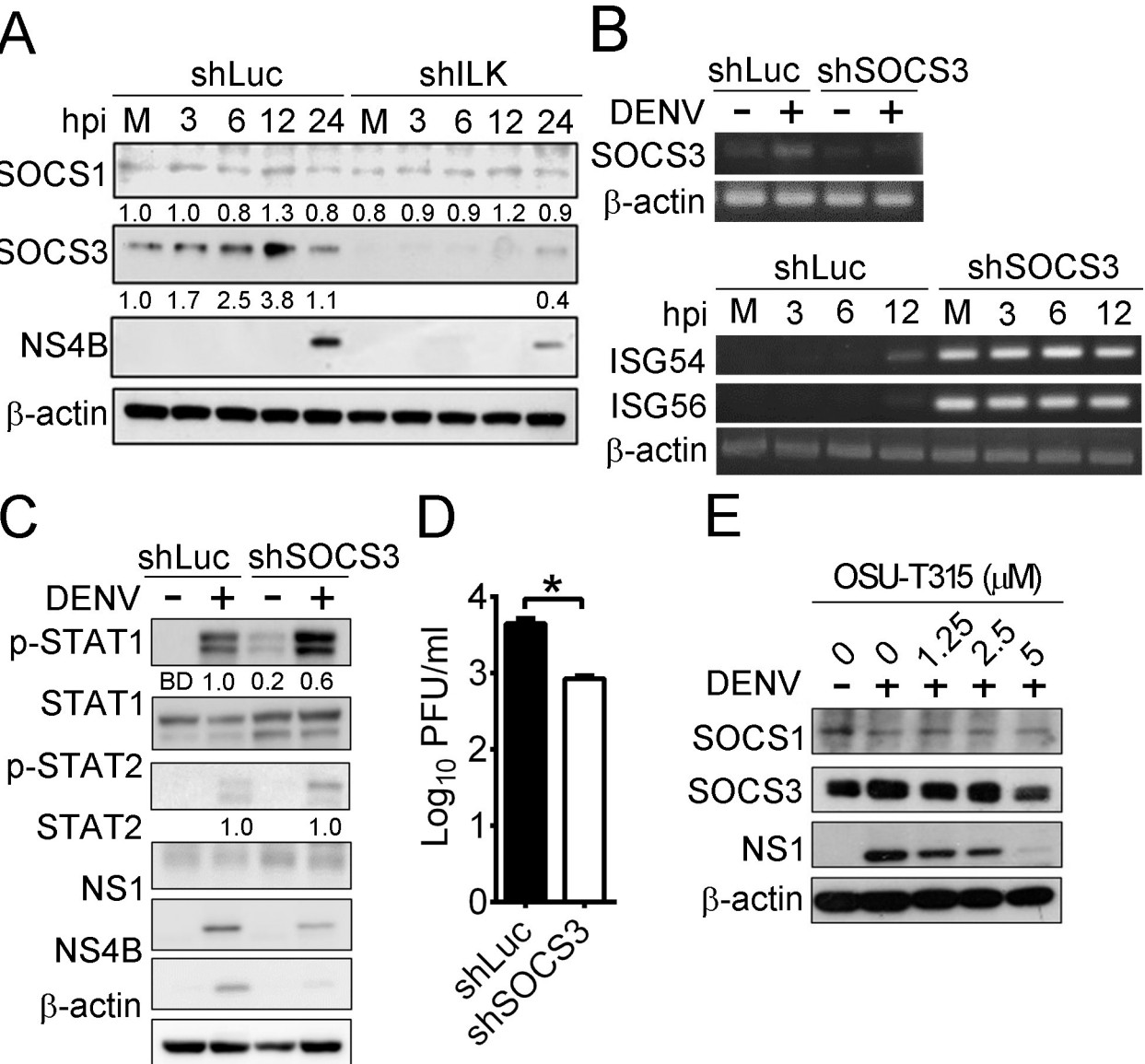

**Fig 4. ILK regulates SOCS3 expression to attenuate IFN signaling.** (**A**) Representative western blots of the indicated proteins in mock- (M) and DENV-infected control (shLuc) or ILK knockdown (shILK) A549 cells at indicated hours post-infection (hpi) are shown. The ratios of SOCS1 or SOCS3 to β-actin relative to mock-infected control cells are shown. (**B-C**) Representative RT-PCR images of the indicated genes (**B**) and western blots of phosphorylated (p-) and indicated proteins (**C**) in control and SOCS3 knockdown (shSOCS3) A549 cells infected with (+) or without (-) DENV at indicated or 24 hpi are shown. The ratios of phosphorylated to total STAT1 and STAT2 relative to infected control cells are shown. (**D**) The viral yields in indicated A549 cells at 24 hpi are shown. Data represent mean + SD (error bars). $^*p < 0.05$. (**E**) Representative western blots of indicated proteins in A549 cells infected with (+) or without (-) DENV and treated with the indicated concentration of OSU-T315 at 24 hpi are shown. BD, below detection.

κB inhibitor, Bay 11–7082, diminished the DENV-induced p65 phosphorylation and SOCS3 and NS4B expression in cells (**Fig 5B**), indicating that ILK regulates SOCS3 expression via activating NF-κB.

A previous study showed that ILK activates NF-κB via Erk in gastric cancer cell lines [40]. In DENV-infected cells, we also found that ILK knockdown attenuated DENV-induced Erk phosphorylation (**Fig 5C**). Treatment with the Erk inhibitor, U0126, suppressed DENV-induced p-Erk, p-IκBα, SOCS3, and NS3 expression in a dose-dependent manner (**Fig 5D**).

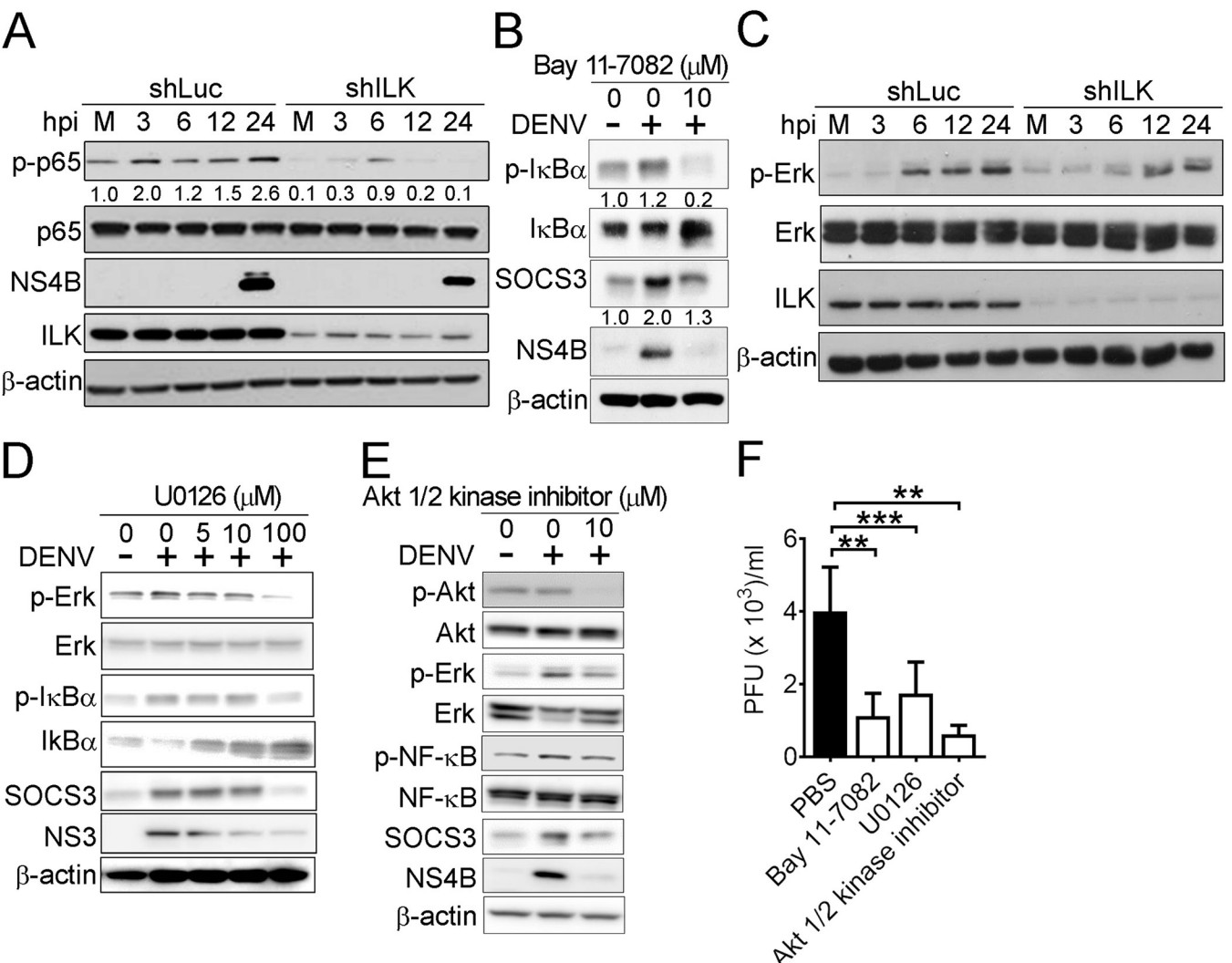

**Fig 5. ILK regulates SOCS3 expression via Akt-Erk-NF-κB pathway.** (**A and C**) Representative western blots of phosphorylated (p-) and indicated proteins in mock- (M) and DENV-infected control (shLuc) and ILK knockdown (shILK) A549 cells at indicated hours post-infection (hpi) are shown. The ratios of phosphorylated to total p65 relative to mock-infected cells are shown. (**B, D, and E**) Representative western blots of indicated proteins in mock-infected (-) or DENV-infected (+) A549 cells treated with indicated concentrations of Bay 11–7082 (**B**), U0126 (**D**), or Akt inhibitor (**E**) at 24 hpi are shown. (**F**) The DENV yields in infected A549 cells treated with 10 μM of indicated inhibitors at 24 hpi are shown. Data represent mean + SD (error bars). $^{**}p < 0.01$, $^{***}p < 0.001$.

Treatment of the Akt1/2 kinase inhibitor reduced DENV-induced p-Erk as well as Erk-NF-κB-driven SOCS3 expression and NS4B expression (**Fig 5E**). Furthermore, DENV yields were reduced in cells treated with inhibitors of NF-κB, Erk, and Akt (**Fig 5F**). This shows that ILK activates the Akt-Erk-NF-κB pathway to increase SOCS3 expression and facilitate DENV infection.

## DENV NS1 and NS3 interact with ILK to attenuate IFN signaling

DENV NS1, NS3 and NS5 have been shown to activate NF-κB pathways [41–43]. We next tested whether these DENV proteins participate in ILK-mediated SOCS3 induction. Ectopic expression of DENV NS1 or NS3, but not NS5, increased the expression levels of p-Akt, p-NF-κB, as well as SOCS3 (**Fig 6A**). Treatment of OSU-T315 attenuated DENV NS1 or NS3 induced SOCS3 (**S8 Fig**). In addition, ectopic expression of DENV NS1 or NS3 reduced IFN-

β-triggered p-STAT1, ISG56, and MxA (**Fig 6B and 6C**). These results indicate that DENV NS1 and NS3 are involved in ILK-mediated IFN signaling inhibition. To understand how DENV NS1 and NS3 affect ILK, we monitored their cellular distribution. The co-localization of ectopic expressed DENV NS1 or NS3, but not NS5, with ILK was clearly observed (**S9A Fig**). The interaction between ectopic expressed DENV NS1 or NS3 with ILK was confirmed by co-immunoprecipitation analysis (**Fig 6D**). In line with these observations, we also found a similar co-localization of DENV NS1 or NS3, but not NS5, with endogenous ILK in DENV-infected A549 cells (**S9B Fig**). The interaction between DENV NS1 or NS3 with endogenous ILK was also further confirmed in DENV-infected A549 cells by co-immunoprecipitation analysis (**Fig 6E**) as well as proximity ligation assay, a powerful tool with high specificity and sensitivity to detect endogenous protein interaction (**S9C Fig**).

ILK contains three domains, including N-terminal ankyrin (ANK) repeats, plekstrin homology (PH) domain and a C-terminal kinase domain, in which ANK and kinase domains are able to bind several cellular proteins such as ILK-associated phosphatase and Akt, respectively, to regulate ILK activity [44]. To understand whether ANK or kinase domain is responsible for interacting with DENV NS1 and NS3, we introduced two ILK constructs ($ILK_{1-170}$ or $ILK_{171-452}$) into NS1 or NS3 expressing A549 cells. By co-immunoprecipitation analysis, we found that only $ILK_{171-452}$ was able to interact with DENV NS1 and NS3, indicating that ILK kinase domain but not ANK domain contributes to this interaction (**Fig 6F**). Furthermore, we also observed the increased Akt recruitment to ILK complex in the presence of DENV NS1 and NS3 (**Fig 6G**). Collectively, these results reveal that DENV NS1 and NS3 interact with ILK to attenuate type 1 IFN signaling.

## Inhibition of ILK reduces viral loads and the mortality rate of DENV2-infected mice

To test whether inhibition of ILK protects hosts against DENV infection, the therapeutic effect of OSU-T315 in DENV-infected mice was examined. 7-day-old C57BL/6 mice were infected with $6 \times 10^5$ PFU of DENV2 via intraperitoneal and intracranial injections. Infected mice were treated with or without 12.5 mg/kg OSU-T315 on the day of infection via intraperitoneal injection. In the control-treated mice, all infected mice displayed neurological symptoms, including hunched back, seizure, and paralysis in both fore- and hind-limbs after 6 days of infection (**Fig 7A**) and succumbed to death (0/8) within 10 days post-infection (dpi) (**Fig 7B**). With OSU-T315 treatment, infected mice exhibited a delayed onset of neurological symptoms, and half of them exhibited little or no neurological symptoms (**Fig 7A**). The survival rate of OSU-T315-treated mice was 50% (4/8) by 21 dpi, which was significantly higher than that in control-treated mice (0/8) (**Fig 7B**), showing that inhibition of ILK protects mice from lethal DENV infection. We further determined the viral loads and NS3 expression in the brains of infected mice on 7 dpi, when virus load reaches a peak. In the brains of OSU-T315-treated mice, the mean viral load (**Fig 7C**) and the NS3 expression (**Fig 7D**) were both reduced compared with those in control-treated mice. Additionally, the levels of TNF-α, as well as IL-6, were significantly lower in the brains of OSU-T315-treated mice than those of control-treated mice on 7 dpi, indicating OSU-T315 attenuates DENV-induced inflammation in mouse brains (**Fig 7E**). Taken together, our findings revealed that inhibition of ILK reduces not only viral loads but also inflammation in infected brains and improves the survival of DENV-infected mice.

## Discussion

In the present study, we revealed that ILK is an enhancing factor for DENV replication (**Fig 7F**). ILK increases the expression of SOCS3 via the Akt-Erk-NF-κB pathway by interacting with DENV NS1 and NS3. Elevated SOCS3 expression antagonizes STAT1/2 activation and

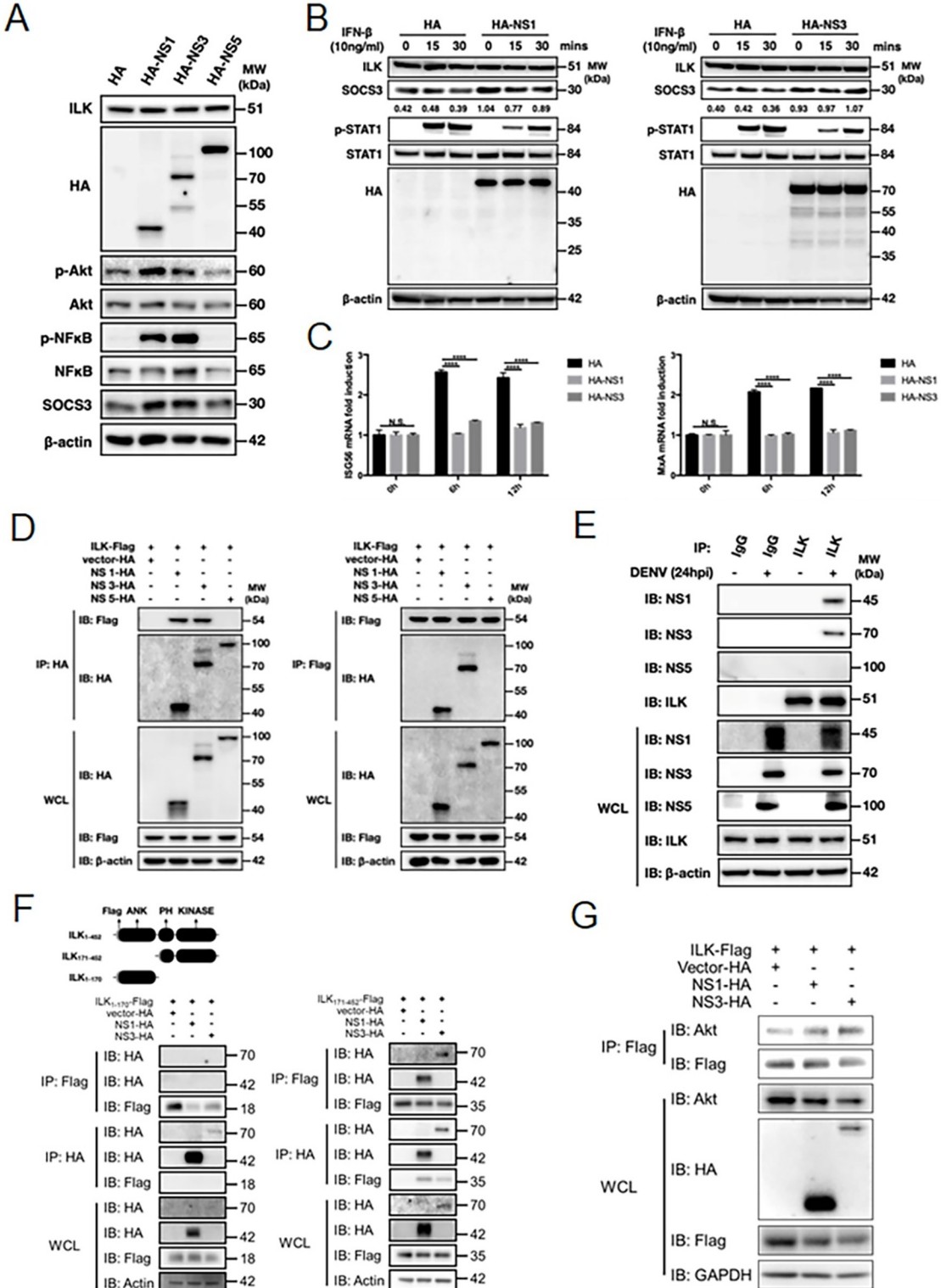

**Fig 6. DENV NS1 and NS3 interact with ILK to attenuate IFN signaling. (A)** Representative western blots of indicated proteins in HA, HA-NS1, HA-NS3, and HA-NS5 expressing A549 cells are shown. **(B and C)** Representative western blots of indicated proteins **(B)** and quantitative RT-PCR of indicated genes **(C)** in HA, HA-NS1, and HA-NS3 expressing A549 cells at indicated hpi are shown. **(D)** A549 cells were transfected with Flag-ILK, HA, HA-NS1, HA-NS3, or HA-NS5 and co-immunoprecipitation performed by anti-HA or anti-Flag antibodies. Representative western blots of indicated proteins are shown. **(E)** Representative western blots

of indicated proteins in control or infected cells after co-immunoprecipitation by anti-ILK antibody are shown. **(F)** A549 cells were transfected with Flag-ILK$_{1-170}$, Flag-ILK$_{171-452}$, HA-NS1 or HA-NS3 and co-immunoprecipitation performed by anti-Flag antibody. Representative western blots of indicated proteins are shown. **(G)** A549 cells were transfected with Flag-ILK, HA, HA-NS1 or HA-NS3 and co-immunoprecipitation performed by anti-Flag antibody. Representative western blots of indicated proteins are shown. The ratios of Akt co-immunoprecipitated with ILK to ILK precipitated by anti-Flag antibody are shown. Data represent mean + SD (error bars). $^{***}p < 0.001$. WCL, whole cell lysate.

inhibits ISG, but not type I and III IFN production. Knocking down ILK or SOCS3 enhances ISG expression and reduces viral yields in cells. Moreover, treatment of the ILK inhibitor, OSU-T315 decreases brain viral loads, disease severity, and mortality of DENV-infected mice.

The integrin β1 and β3 are shown to promote DENV binding and entry into cells [10,45]. Although ILK is bound to and activated by integrin β1 and β3, the level of DENV binding to cells is not affected by ILK. Moreover, in Huh7 cells with a luciferase-expressing DENV replicon, ILK knockdown attenuated luciferase activity and viral protein expression. These results indicate an enhancing role of ILK in DENV replication after virus entry. During replication, DENV activates PI3K to facilitate autophagy and activates Akt to prevent apoptosis of infected cells [13]. Nonetheless, there was no difference in the levels of LC3 I/II conversion and cleaved caspase-3 between wild-type and ILK knockdown cells after DENV infection. These results show that ILK-mediated enhancement in DENV infection is independent of autophagy or apoptosis.

DENV has been shown to disrupt IFN and ISG induction via increasing SOCS1 and SOCS3 expression [22,23], but the detailed mechanism is still lacking. In this study, we revealed that ILK interacts with DENV NS1 and NS3 to increase SOCS3 expression and subvert antiviral IFN signaling. This SOCS3 induction is mediated by ILK-Akt-Erk-NF-κB signaling. Pharmaceutical or genetic inhibition of ILK decreases SOCS3 expression in both mock- and DENV-infected cells. Notably, knocking down ATP6V0C, an irrelevant DENV host-dependency factor [46], reduced DENV-induced p-p65 as well as NS1 and NS3 protein levels and SOCS3 protein levels, while ILK levels remained unchanged (**S10 Fig**). It might partly result from decreased NS1/NS3 and ILK interaction, as we suggested, or other DENV-activated signalings that have not been identified, such as pleiotropic effects resulting from lower replication of DENV. Further studies are needed to address the issue. We found that SOCS3 mRNA levels were only 1.4- to 1.3-fold higher in control cells than in ILK knockdown cells after DENV infection (**S4E Fig**). However, the protein levels of SOCS3 were dramatically reduced in ILK knockdown cells with or without DENV infection compared to control cells (**Fig 4A**). A recent study revealed that ILK is able to increase oncogenic protein synthesis by positively regulating translation via AKT/mTOR/eIF4E pathways [47]. According to these observations, we suggest that ILK might upregulate both SOCS3 RNA expression and translation as well to support DENV infection. As one of ILK kinase substrate, Akt is able to be phosphorylated at Ser 473 by ILK for its activation. Accordingly, Akt has been found to bind to ILK via Rictor to be activated [48]. Our results showed that DENV NS1 and NS3 bind to the kinase domain of ILK, and also showed an increase of Akt recruitment to this complex. The involvement of Rictor in forming this DENV NS-ILK complex is worthy of further investigation. Notably, several studies have pointed out that dengue viral non-structure proteins may antagonize IFN-signaling. For example, DENV NS2A, NS4A, and NS4B proteins inhibit STAT1 phosphorylation [15,49], and NS1 and NS2B3 activate NF-κB-dependent transcription [41,50]. DENV NS5 co-opts ubiquitin protein ligase E3 component n-recognin 4 (UBR4) to degrade STAT2 and antagonize type 1 IFN signaling [51]. Our findings here explored the new roles of DENV NS1 and NS3 in triggering SOCS3-mediated IFN signaling inhibition via ILK. Interestingly, previous studies revealed that overexpression of DENV NS1 and NS3 is able to induce IL-6 and IL-10 production, respectively [52,53]. Both IL-6 and IL-10 transduce a STAT3-dependent signal pathway,

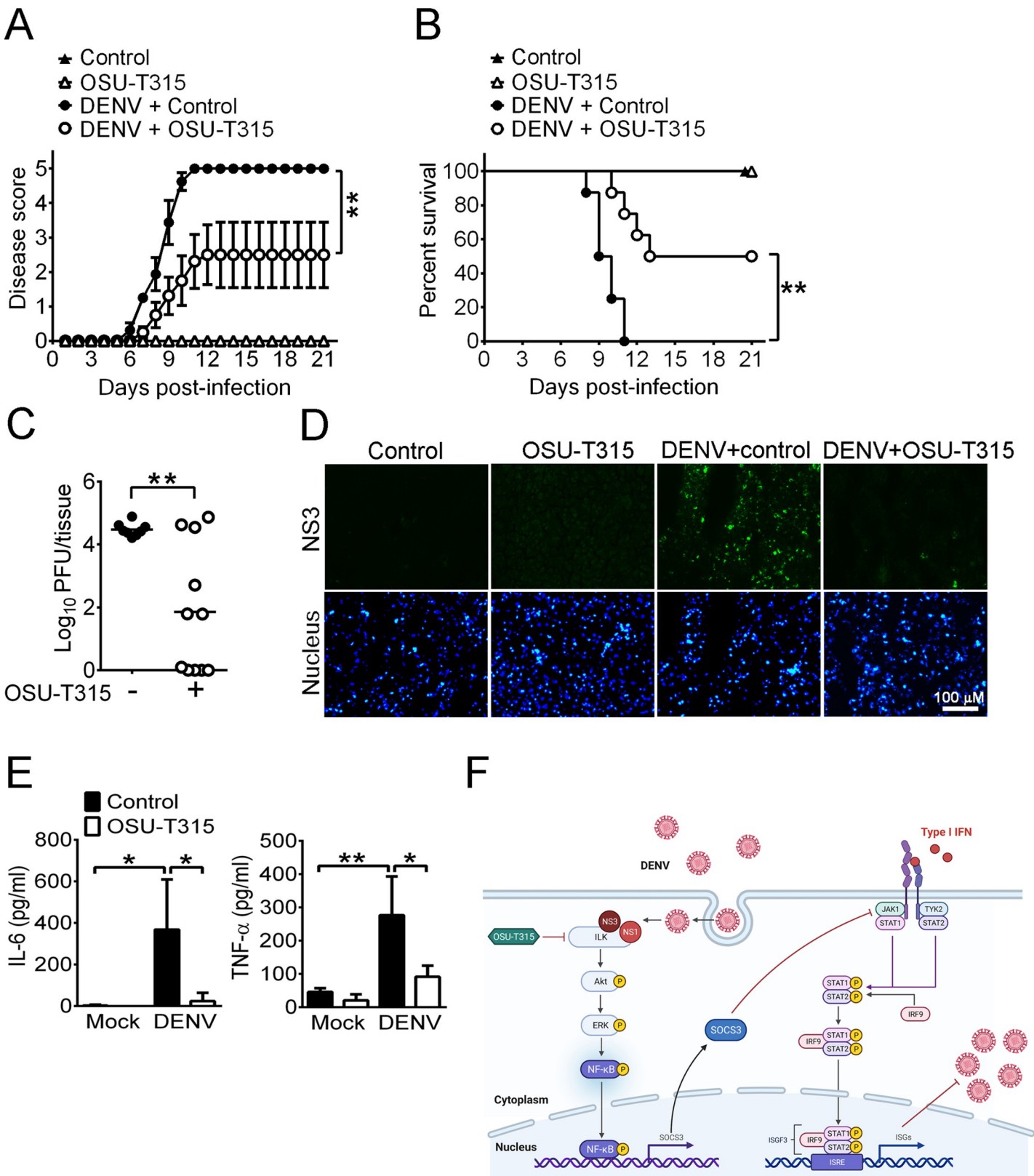

**Fig 7. The inhibition of ILK reduces viral loads in the brains and mortality rate of DENV2-infected mice. (A, B, C, and E)** The 7-day-old C57BL/6 mice were mock-infected or infected with $6 \times 10^5$ plaque-forming unit (PFU) of DENV and treated without (control or -) or with 12.5 mg/kg OSU-T315. The disease scores (**A**), survival

rate (**B**), viral loads in the mouse brains on day 7 post-infection (**C**), and levels of indicated proteins in the mouse sera day 7 post-infection (**E**) are shown. (**D**) The images of cortex regions stained with anti-NS3 (green) antibody from brain sections of mice mock- or DENV-infected and treated without (control) or with OSU-T315 on day 7 post-infection are shown. Nuclei were counterstained with DAPI. (**F**) The schematic diagram of mechanism mediated by ILK to promote DENV infection. This figure is created with Biorender.com. Data represent the mean ± SD (error bars) in panel A and the mean + SD (error bars) in panel E. Each symbol on the scattergram represents an individual sample. The black horizontal line represents the mean value for each group in panel C. $^{*}p < 0.01$ and $^{**}p < 0.001$ of 5 samples per data points via two-way ANOVA with the appropriate post-hoc test for multiple comparisons in panel A and E, via log-rank test in panel B, and via Student's t test in panel C.

which has been shown to negatively regulate type-I IFN responses [54]. Thus, DENV NS1 and NS3 might also antagonize IFN signaling by activation of STAT3-mediated actions. Overall, these observations underscore the conclusion that DENV develops multiple strategies to subvert the cellular antiviral response.

Besides DENV, ILK has also been shown to support the infection of coxsackievirus B3 and the hepatitis C virus [30,31]. Especially in hepatitis C virus-infected cells, ILK activation also attenuates ISG expression. Furthermore, infection of hepatitis C virus [55], influenza A virus [39,56], respiratory syncytial virus [57], human immunodeficiency virus [58], and herpes viruses [59,60], induce SOCS3 expression to antagonize STAT1/2 activation, implying SOCS3 can be a therapeutic target for the development of broad-spectrum antiviral drugs. Moreover, it is essential to determine if the ILK-Akt-Erk-NF-κB pathway we identified in DENV-infected cells is responsible for the induction of SOCS3 in cells infected with other viruses.

To study if ILK inhibitor, OSU-T315, protects the host from DENV infection, we chose the mouse model in which neonatal (7-day-old), immunocompetent mice were infected with DENV2 via intracranial and intraperitoneal injections [61,62]. DENV infection in immunocompetent mice fails to induce overt signs of diseases. Although the immunocompromised mice, such as mice with deficiency in type I and II IFN responses, support robust DENV replication, and display severe signs of disease [63–65], they are unsuitable for our study due to the lack of intact IFN responses. In contrast, in the immunocompetent neonatal mice, DENV replicates in the brains, induces neurological symptoms, such as seizure and paralysis, and results in death after infection, which makes it a feasible model for us to test the efficacy of blocking ILK as a therapeutic strategy. Furthermore, the central nervous system involvement is now considered as a criterion for severe dengue according to the World Health Organization classification [66], despite it having been thought an uncommon complication in the past two decades. Using this model, we demonstrated that inhibition of ILK by OSU-T315 improves the survival rate of DENV-infected mice with decreased viral loads in the brains, and reduced disease severity. These results suggest that ILK might serve as a promising therapeutic target for treating dengue.

Since the levels of TNF-α, IL-6, and SOCS3 in the blood are higher in dengue-infected patients with hemorrhagic fever or shock (severe dengue) than those in patients with fever only [24], it suggests that overt inflammatory responses are detrimental to DENV-infected hosts. The observation that treatment of anti-TNF serum protects mice from lethal DENV infection [67] supports this hypothesis. We found that the levels of TNF-α and IL-6 in infected mice were lower in OSU-T315-treated mice than in control mice, implying that inhibition of ILK reduces not only viral titers but also inflammatory cytokines. Others also demonstrate that inhibiting ILK abrogates NLRP3-mediated IL-1β production [68], that ILK knockdown reduces TNF-α production via NF-κB [69], and that ILK knockout in intestinal epithelial or myeloid cells reduces DSS-induced colitis in mice [70,71]. Further studies are needed to determine the cell types and mechanisms of ILK-induced inflammatory responses. Also, it would be essential to examine if ILK expression is elevated in DENV-infected patients with comorbidities who are susceptible to develop severe dengue.

In summary, we show that inhibition of ILK is a novel and effective therapeutic strategy against DENV replication in cells, and DENV lethality in mice, by reducing viral titers as well as inflammatory cytokines.

## Materials and methods

### Ethics statement

All mouse experiment protocols were approved by the Institutional Animal Care and Use Committee of National Cheng Kung University (IACUC numbers 104062 and 105022). The C57BL/6 mice were maintained under a specific-pathogen-free condition in the Laboratory Animal Center of our college.

### Cells and viruses

The C6/36 (an *Aedes albopictus* cell line), BHK-21 (a baby hamster cell line), A549 (a human lung carcinoma cell line), Huh7 (a human hepatocarcinoma cell line), and 293T (a human embryonic kidney cell line) were maintained in the medium according to the instructions of the American Type Culture Collection. Huh7 cells harboring DENV2 replicon with luciferase reporter were established previously [34] and cultured in DMEM with 10% FBS and selected by 500 μg/ml G418. A549 cells were used for most experiments unless specified. The shRNA clones targeting ILK 3'UTR (TRCN0000000968) or CDS (TRCN0000000969), SOCS3 (TRCN0000057073), LacZ (TRCN0000072224), and luciferase (TRCN0000072244) were obtained from the National RNAi Core Facility, Institute of Molecular Biology and Genomic Research Center, Academia Sinica, Taipei, Taiwan. The shRNA targeting ILK 3'UTR was used to establish ILK knockdown cells otherwise specified. Lentiviruses carrying specific shRNA were propagated in 293T cells transfecting with 1.5 μg pCMVΔ8.91, 0.5 μg pMD.G, and 1 μg pLKO-puro with polyethylenimine (Polysciences). To establish polyclonal cells with specific shRNA, cells were transduced with lentiviruses in medium with 8 mg/ml polybrene (Sigma-Aldrich) for 48 hours and selected in medium containing puromycin (Sigma-Aldrich) for 4 days. Taiwan clinical isolates of DENV serotype 1 (8700828), 2 (PL046), 3 (8700829), and 4 (8700544) were propagated in C6/36 cells and titrated on monolayers of BHK-21 cells. DENV2 was used for most of the experiments unless specified. The pcDNA3.1-2HA plasmids (encoding HA tag at both ends of multiple cloning sites) expressing DENV2 (PL046) NS1 (HA-NS1), NS3 (HA-NS3), and NS5 (HA-NS5) were established previously [34]. The pcDNA3.1(+) plasmids expressing full-length ILK, $ILK_{1-170}$ and $ILK_{171-452}$ tagged with Flag at N-terminus were purchased from Leadgene Biomedical, Inc.

### Infection and treatment of mice

The 7-day-old C57BL/6 mice were infected with a total of $6 \times 10^5$ plaque-forming unit (PFU) per mouse of DENV2 via intracerebral ($1.5 \times 10^5$ PFU) and intraperitoneal ($4.5 \times 10^5$ PFU) injections [61] and treated either with or without 12.5 mg/kg of OSU-T315 via intraperitoneal injection on the day of infection. Mice were monitored daily for survival rates and given disease scores as follows: 0, healthy; 1, hunched back; 2, seizure or wild rushing and jumping; 3, weakness or partial paralysis in limbs; 4, complete paralysis in hind and forelimbs; 5, moribund (score it as 5 rather than 4) or death. In separate experiments, the mouse brains were harvested on indicated days after infection to determine viral loads in the brain via plaque assay or for immunofluorescence staining.

## Western blot analysis and immunoprecipitation

Cells were either mock-infected or infected with DENV at a multiplicity of infection (MOI) of 1–2 for 2 hours at 37˚C. After infection, cells were washed, incubated at 37˚C in the medium with or without indicated concentrations of OSU-T315 (MedChemExpress), Bay 11–7082 (Tocris Bioscience), U0126 (Cell Signaling Technology Inc.), or Akt1/2 kinase inhibitor (Sigma-Aldrich), and harvested at the indicated hpi in lysis buffer (Cell Signaling Technology). Cell lysates were subjected to western blot analysis using primary antibodies against Akt with phosphorylation (p-) on Ser473, Akt, ILK, p701-STAT1, STAT1 (Epitomics), p690-STAT2, STAT2 (Abcam), p536-p65, p65, SOCS3 (Proteintech), p32-IκBα, IκBα, p202/204-Erk, Erk, NS1 (clone 33-D2 kindly provided by Dr. Trai-Ming Yeh), NS3 (GeneTex), NS4B (GeneTex), prM (clone 155–49), and β-actin (Abcam) and HRP-conjugated secondary antibodies. Most primary antibodies were obtained from Cell Signaling Technology unless specified. Protein bands were detected using an enhanced chemiluminescence substrate kit (PerkinElmer). The intensity of protein bands was measured via ImageJ software. For immunoprecipitation analysis, cell lysates were immunoprecipitated by anti-HA or anti-Flag antibodies followed by immunoblotting as described above. All western blots are representative images from 2 independent experiments.

## Immunofluorescence staining

DENV-infected A549 cells were fixed, permeabilized, and stained with primary antibody against double-stranded RNA (SCICONS, clone J2). The mouse brains were harvested and embedded in OCT compound. The longitudinal mouse brain sections (10 μm) were prepared by the sliding freezing microtome, fixed in 3.7% formaldehyde, incubated with cold-acetone, and stained with primary antibody against NS3. Alexa Fluro 488-conjugated secondary antibodies were used to visualize the signal in mouse cortex regions under a fluorescence microscope. Nuclei were stained with DAPI.

## Cell viability assay

The culture supernatants of cells treated with or without the indicated concentrations of OSU-315 for 24 hours were collected to determine the activity of released LDH using a cytotoxicity detection kit (Roche) according to the manufacturer's instructions. The cell viability of cells treated with 0.05% Triton-X 100 was set as 0%.

## RT-PCR and RT-quantitative PCR

Total RNA was prepared via a rNeasy RNA Mini Kit (Qiagen) according to the manufacturer's instructions. The cDNA was reversely transcribed from 1 μg total RNA by random primers and subjected to quantitative real-time PCR reactions with a LightCycler FastStart DNA Master PLUS SYBR Green I kit (Roche), and primers used were as follows: MxA forward ACCAC AGAGG CTCTCAGCAT and reverse CTCAGCTGGTCCTGGATCTC; β-actin forward AAGGAGAAGCTGTGCTACGTCGC and reverse AGACAGCACTGTGTTGGCGTACA. The relative expression of a specific gene was calculated as $2^{-\Delta\Delta CT}$. For RT-PCR, equal amounts of cDNA were subjected to PCR reactions with primers as follows: ISG54 forward ATGTGCA ACCTACTGGCCTAT and reverse TGAGAGTCGG CCCATGTGATA; ISG56 forward GGGCAGACTGGCAGAAGC and reverse TATAGCGGAAGGGATTTGAAAGC; and β-actin forward AAGGAGAAGCTGTGCTACGTCGC and reverse AGACAGCACTGTGT TGGCGTACA. PCR products were separated in 2% agarose DNA gel.

## ELISA

The culture supernatants of cells, either mock-infected or infected with DENV at 24 hpi, and the mouse serum, were collected and subjected to tests to determine the levels of IFN-β in the culture supernatants, and TNF-α and IL-6 in the mouse serum, using ELISA kits (R&D Systems) according to the manufacturer's instructions.

## Statistics

Data are expressed as mean + or ± standard deviation (SD). For statistical comparison, the luciferase activity or cell viability was analyzed via one-way ANOVA followed by Dunnett's post hoc test for multiple comparisons, while the MxA mRNA levels, cytokine levels in the mouse serum, and disease scores were analyzed via two-way ANOVA followed by Tukey post hoc test for multiple comparisons. The viral titers in cells or mouse brains were analyzed using Student's *t* test, and the Kaplan-Miller survival curves were analyzed via log-rank test. Analyses were performed using GraphPad Prism 4 software (GraphPad Software). All *p* values are for two-tailed significance tests. A *p* value of $< 0.05$ was considered statistically significant.

## Supporting information

**S1 Text. Supporting information of Materials and Methods.**
(DOCX)

**S1 Fig. The effects of DENV infection and ILK blockage on Akt Ser473 phosphorylation and cell viability.** (**A**) Representative western blots of phosphorylated (p-) Akt at Ser473 and indicated proteins in mock- (M) and DENV-infected A549 cells at indicated hours post-infection (hpi) are shown. The ratios of phosphorylated to total Akt relative to mock-infected cells are shown. (**B**) The viability of control (shLuc) and ILK knockdown (shILK) A549 cells with mock-infected, infected with DENV at indicated hpi, or treating with 0.05% Triton X-100 for 24 hours are shown. The viability of mock-infected control cells was set as 100%. (**C and D**) Representative western blots of indicated proteins (**C**) and cell viability (**D**) of A549 cells treated with indicated concentration of OSU-T315 for 24 hours are shown. The ratios of phosphorylated to total Akt relative to cells treated with 0 μM OSU-T315 are shown. The viability of cells treated with 0 μM was set as 100%. Data represent means + SD (error bars) in bar graphs.
(TIF)

**S2 Fig. ILK does not play a role in DENV binding.** The control (shLuc) or ILK knockdown (shILK) A549 cells were infected with DENV at MOI of 25 at 4˚C for 2 hours, washed, and stained with anti-envelope (E) antibody. (**A**) The representative histograms of mock-infected and DENV-infected control and ILK knockdown cells are shown. (**B**) The percentages of E-positive cells are shown. Data represent means + SD (error bars).
(TIF)

**S3 Fig. ILK fails to affect DENV-induced autophagy and apoptosis in cells.** The representative western blots of indicated proteins in mock- (M) and DENV-infected control (shLuc) or ILK knockdown (shILK) A549 cells at indicated hours post-infection (hpi) are shown.
(TIF)

**S4 Fig. ILK increases SOCS3 and reduces ISG, but not type I and III IFN, expression.** The control (shLuc) or ILK knockdown (shILK) A549 cells were mock- (M) or DENV-infected and collected at indicated hours post-infection (hpi) to determine the mRNA levels type I (α and β) IFN (**A**), type III (λ) IFN (**B**), ISGs, including MxA, ISG56, and ISG54 (**C and D**),

SOCS1, and SOCS3 (**E**). The representative RT-PCR images of indicated genes are shown, and the ratios of SOCS3 to β-actin relative to control cells at 12 hpi are shown. The SOCS1 and SOCS3 mRNA levels determined by real-time PCR are shown (**E**, right panels). Cells treated with 10 μg/ml poly(I:C) for 24 hours served as positive controls. $^{**}p < 0.01$.
(TIF)

**S5 Fig. ILK regulates SOCS3 expression to suppress IFN-β signaling.** The representative western blots of indicated proteins in control (shLuc) or ILK knockdown (shILK) A549 cells treated with IFN-β (50 ng/ml) for indicated minutes (min) are shown.
(TIF)

**S6 Fig. DENV replication is essential for SOCS3 induction.** (**A**) A549 cells were infected with DENV or UV-inactivated DENV (UV-DENV) at MOI of 25 at 4°C for 2 hours, washed, and stained with anti-envelope (E) antibody. The percentages of E-positive cells analyzed by flow cytometry are shown. (**B**) The representative western blots of the indicated proteins in the A549 cells infected with DENV or UV-DENV for 24 hours are shown. The ratios of SOCS3 to β-actin relative to mock-infected cells are shown. Data represent means + SD (error bars) in panel A. N.S, not significant.
(TIF)

**S7 Fig. ILK increases SOCS3 expression to inhibit IFN signaling in DENV-infected U937 cells.** (**A**) The representative western blots of phosphorylated STAT1 (p-STAT1) and indicated proteins in control (shLuc) or ILK knockdown (shILK) cells infected with DENV at indicated hour post-infection (hpi) are shown. (**B**) The DENV yields in control and ILK knockdown cells at 24 hpi are shown. Data represent mean + SD (error bars).
(TIF)

**S8 Fig. OSU-T315 attenuates DENV NS1 or NS3 induced SOCS3.** The representative western blots of indicated proteins in HA, HA-NS1 and HA-NS3 expressing A549 cells treated with OSU-T315 at indicated concentrations are shown. The ratios of SOCS3 to β-actin relative to HA-expressing cells without OSU-T315 treatment are shown.
(TIF)

**S9 Fig. DENV NS1 and NS3 co-localize with ILK. (A)** The representative confocal images of HA-NS1, HA-NS3 and HA-NS5 expressing cells stained with ILK (green) or HA (red) are shown. **(B)** The representative confocal images of mock or DENV-infected cells stained with ILK (red) or NS proteins (green) are shown. **(C)** The representative images of proximity ligation assay in mock or DENV-infected cells are shown.
(TIF)

**S10 Fig. Knockdown of ATP6V0C attenuates DENV-induced p-p65, SOCS3 and viral protein expression. (A)** ATP6V0C mRNA levels in A549 cells transfected with scramble or ATP6V0C-specific siRNA determined by qRT-PCR are shown. **(B)** The phosphorylated (p-) p65 and indicated proteins in the mock- or DENV-infected A549 cells transfected scramble control siRNA or siRNA targeting ILK or ATP6V0C at indicated hpi are shown. The ratios of phosphorylated to total p65 relative to mock-infected cells transfected with scramble siRNA at 0 hpi are shown. **(C)** The indicated A549 cells were infected with DENV (MOI of 2) for 0 and 24 hours. The NS1, NS3, SOCS3, and GAPDH protein expression in the indicated A549 cells are shown. The ratios of SOCS3 to GAPDH relative to mock-infected scramble control cells and the ratio of NS1 and NS3 to GAPDH relative to DENV-infected scramble control cells are shown.
(TIF)

## Acknowledgments

We thank the technical services provided by the "Bioimaging Core Facility of the National Core Facility for Biopharmaceuticals, Ministry of Science and Technology, Taiwan" and the support from the "Core Research Laboratory, College of Medicine, National Cheng Kung University".

## Author Contributions

**Conceptualization:** Chih-Peng Chang.

**Data curation:** Yi-Sheng Kao, Li-Chiu Wang, Po-Chun Chang, Heng-Ming Lin.

**Formal analysis:** Li-Chiu Wang, Chih-Peng Chang.

**Funding acquisition:** Chih-Peng Chang.

**Investigation:** Yi-Sheng Kao, Li-Chiu Wang, Po-Chun Chang, Heng-Ming Lin, Chien-Chou Chu, Bo-Cheng Zhang.

**Methodology:** Yi-Sheng Kao, Li-Chiu Wang, Po-Chun Chang, Heng-Ming Lin, Yee-Shin Lin, Chia-Yi Yu, Chien-Chin Chen, Chiou-Feng Lin, Trai-Ming Yeh, Shu-Wen Wan, Chien-Chou Chu, Bo-Cheng Zhang.

**Project administration:** Chih-Peng Chang.

**Resources:** Yee-Shin Lin, Chia-Yi Yu, Trai-Ming Yeh, Shu-Wen Wan, Jen-Ren Wang, Tzong-Shiann Ho.

**Supervision:** Chih-Peng Chang.

**Validation:** Yi-Sheng Kao, Li-Chiu Wang, Po-Chun Chang, Heng-Ming Lin, Chien-Chou Chu, Bo-Cheng Zhang.

**Writing – original draft:** Yi-Sheng Kao, Li-Chiu Wang, Chih-Peng Chang.

**Writing – review & editing:** Yi-Sheng Kao, Li-Chiu Wang, Po-Chun Chang, Heng-Ming Lin, Yee-Shin Lin, Chia-Yi Yu, Chien-Chin Chen, Chiou-Feng Lin, Trai-Ming Yeh, Shu-Wen Wan, Jen-Ren Wang, Tzong-Shiann Ho, Chien-Chou Chu, Bo-Cheng Zhang, Chih-Peng Chang.

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
