## [Decision Letter · Decision Letter 0]

21 Jul 2022

Dear Dr. Chang,

Thank you very much for submitting your manuscript "Negative regulation of type I interferon signaling by integrin-linked kinase permits dengue virus replication" for consideration at PLOS Pathogens. As with all papers reviewed by the journal, your manuscript was reviewed by members of the editorial board and by several independent reviewers. In light of the reviews (below this email), we would like to invite the resubmission of a significantly-revised version that takes into account the reviewers' comments.

We cannot make any decision about publication until we have seen the revised manuscript and your response to the reviewers' comments. Your revised manuscript is also likely to be sent to reviewers for further evaluation.

Sincerely,

Kellie A. Jurado

Associate Editor

PLOS Pathogens

Stacy Horner

Section Editor

PLOS Pathogens

Kasturi Haldar

Editor-in-Chief

PLOS Pathogens

orcid.org/0000-0001-5065-158X

Michael Malim

Editor-in-Chief

PLOS Pathogens

orcid.org/0000-0002-7699-2064

Reviewer's Responses to Questions

**Part I - Summary**

Reviewer #1: In this study, Kao and colleagues describe a negative regulation of type on interferon signalling during Dengue infection. The authors claim that this phenomena is mediated by integral-linked kinase, NS1 and 3 and activation of NF-kB, that culminates in the transcription of SOCS3. To demonstrate their points the authors use celular and animal models as well as RNA interference and pharmacological tools. Most of the claims are directly supported by the experiments.

Reviewer #2: This is an interesting study. It uncovers a new pro-viral host factor, ILK, that can enhance dengue virus replication. It also uncovers a new dengue virus virulence factor, as NS1 and NS3 are somehow involved in the pro-viral activity of ILK. The data represented in this study is quite convincing.

Reviewer #3: This manuscript by Kao and colleagues reports ILK, an integrin-like kinase, as a new host-dependency factor positively regulating DENV replication via restriction of typeI IFN response. Using a combination of genetic and chemical inhibitors of different downstream effectors of this pathway, the authors present evidence supporting activation of the AKT-ERK-NfKB-SOCS3 via NS1 and NS3, ultimately triggering dephosphorylation of STAT1/2 and inhibition of IFN response.

Furthermore, a small molecule inhibiting ILK1 in an immunocompetent model of DENV pathogenesis reduces viral load and disease severity in mice, suggesting that ILK might serve as potential therapeutic target for DENV infections.

The manuscript is well-written and present a number of interesting and novel observations, including noteworthy animal experiments with small molecule inhibitors as well as convicing data supporting the interaction of ILK with NS1 and NS3 and its functional relevance for DENV replication in vivo and in vitro. However, an important limitation of the study derive from the large use of WB using phospho-specific antibodies as main readout to mechanistically link ILK to type I IFN as well as the use of inhibitors whose cytotoxicity appears not tested. These studies are often performed in conditions of reduced viral replication (due to ILK inhibition) making difficult to rule out whether downstream effects are linked to ILK activity or pleiotropically result from reduced viral replication (independent of ILK). Altogether, the study would greatly benefit from alternative and complementary validations to corroborate the proposed mechanistic links (see detailed comments below).

**Part II – Major Issues: Key Experiments Required for Acceptance**

Reviewer #1: -Can the authors please comment on the fact that in figure 4 the authors show that there is upregulation of ISG54 and 56 in the absence of infection when SOCS3 is knocked down (4D lower panel, 0 h.p.i.). Does this impact the overall conclusion?

-In figure 5 and the overall conclusions of the paper, the authors imply that the NF-kB-mediated transcription of SOCS weights more on the infection that the pro-inflammatory genes. To support this the authors should measure virus growth (PFU/ml) in experiments (similar to) 5B, D and E.

-Can the results found in figure 6C be explained by the other immune modulatory functions already described for NS1 and NS3? The authors should include this in the discussion.

Reviewer #2: 1) The figure captions should be a bit more descriptive, such that a reader can get a good understanding of the experiments performed without referring to the main body of text. Figure 3 is a positive example. The caption for figure 3 makes it clear that all the images represent experiments performed with the A549 cell line. However, for figure 1, the A549 cell line is mentioned for some of the images but not the others. The figure 5 and figure S5 captions do not even mention what cell line is involved.

2) The shRNA knockdown cell lines are used for most of the experiments in this study. The authors should include more details about the shRNA cell lines. Can the authors clarify the technical details of how the shILK knockdown cell lines are established? Was it used as a polyclonal pool, or were clonal cell lines selected after antibiotic selection?

3) For Huh-7, there are two knockdown cell lines: shILK-1 and shILK-2. Which was the shRNA sequence used to generate the knockdown cell lines for U937 and A549?

4) Can the authors elaborate on the shLuc that was used as a control shRNA? Does it target firefly luciferase? If so, it may not be an appropriate control for the replicon since it is expressing firefly luciferase. Wouldn’t the firefly luciferase RNA sequence in the replicon genome be targeted by the shLuc?

5) Authors should elaborate on some of the abbreviations used in the text and manuscript. For example, in figure 6, the abbreviation WCL should be defined in the caption as whole cell lysate

Reviewer #3: 1. One of the main limitations of the study is its exclusive reliance on WB using phospho-specific antibodies to derive mechanistic links between up- and down-stream effectors along the ILK-AKT-NFkB-SOCS3 axis. Most of these experiments rely on minor differences, and appear to be based on individual experiments. Were these experiments carried multiple times? (figure legends only state “representative blot shown”). Importantly, specificity and toxicity of the shRNA and small molecule inhibitors used should be validated (i.e. by rescue experiments using overexpression constructs) by alternative methods as most of the conclusions of the study derive from these inhibitors and WB read-outs (specific examples below).

- Fig 1A: levels of pAKT are increasing over time, both in control (shLuc) and ILK k.d. cells. The relative differences seems rather driven by overall lower protein amounts in kd cells (see ß-actin levels, lines 8-12 compared to 1-7). It would be important to confirm this observation in normalized WB, and show the corresponding cell viability of shILK-treated cells.

- Fig 1B. Are the concentrations at which the inhibitor reduces viral proteins levels cytotoxic? It would be important to show cell viability under the conditions used in this assay, as well as Akt phosphorylation levels upon inhibitor treatment to support its specificity.

- Fig 1B vs Fig 2E: why the levels of NS4B or NS3 are mildly affected with i.e. 2.5 uM of OSU-T315, while in identical conditions in a different panel (Fig. 2E) appear almost completely abrogated?

- Fig 4B (lines 205-208): SOCS3 levels appear not upregulated by DENV infection when comparing i.e. 24h vs 0h post infection, while transcripts levels are comparable in both shLuc and shILK cells. Importantly shILK-mediated silencing does not significantly affect RNA levels of either SOCS1 or 3 in the agarose gel presented (Fig 4A). It would be important to use more quantitative methods to assess its levels rather than DNA electrophoresis (i.e. qRT-PCR). Similar concerns apply to S6Fig (“SOCS3 expression was abrogated”), which should support dependencies of these effects on active RNA replication (however differences in SOCS3 expression appear negligible).

2. The mechanistic link of ILK to SOCS3 activity via NfKB axis mostly rely on virus-infections experiments in which viral replication is inhibited by silencing or drug treatment (see Lines 214-215; Fig. 4; Fig 5 D, E, etc..). However, to conclude that ILK is directly responsible for DENV-induced modulation of SOCS3 expression, it would be important to include controls in which similar reduction in virus replication is achieved (i.e. silencing of an unrelated host-depency factor such as ATP6V0C subunits).

Alternatively, is not possible to rule out pleiotropic effects resulting from lower replication of DENV (upon inhibition of ILK activity or any other host-dependency factor) which would equally result in reduced SOCS3 levels. Similarly, any other conclusion related to the interdependency of ILK and downstream IFN or NFkB pathway might be due to pleiotropic effects deriving from reduced viral replication (i.e. shILK reduces DENV replication, therefore is not surprising to observe slightly lower pERK levels in these conditions) (Fig. 5C).

Notably, experiments which are independent from active viral replication (i.e. ectopic expression of NS1 or NS3 followed by treatment with t315 inhibitor) display negligible effects on SOCS3 abundance upon inhibitor treatment (and protein expression as well)). i.e. Fig. S8.

3. While the interaction between NS1 and NS3 with ILK is convincingly supported by co-IP-WB and PLA experiments, the increased recruitment of Akt to ILK upon NS1 or NS3 expression is not supported by the data presented in the manuscript.

In fact, despite NS1 and NS3 overexpression significantly reduce ISG56 levels (Fig. 6C), AKT-ILK stochiometry appear unaffected by NS1 or NS3 expression (see Akt levels comparable in all lanes, Fig.6G), therefore hampering the conclusion that NS1 and NS3 interact with ILK to attenuate type1 IFN responses.

Similarly, it would be important to overexpress an unrelated ER soluble protein to ensure that increased pAkt and NFkB levels are due to NS1 and NS3 and not deriving from overexpression artefacts (NS5 is expressed in the nucleus and cytoplasm and might actually counteract those pathways in directly) (Fig 6A)

4. Lines 221-222 (Fig 4E): “Furthermore, SOCS3 knockdown enhanced p-STAT1/2 and suppressed NS1 and NS4B expression in infected cells (Fig 4E)”. This blot actually shows upregulation of pSTAT1 levels in both shLuc and shSOCS3 silenced cells, suggesting that hyperphosphorylation of STAT1 is dependent on viral infection but SOCS3-independent..

Lines 230-232 (Fig 5A) “Our data showed that DENV infection increased the phosphorylation of p65 on Ser536 (p-p65) in control cells while ILK knockdown abrogated this p65 phosphorylation”. In the blot presented, p-p65 levels appear comparable (left top blot; pp65 levels i.e. at 3h and 25h.p.i., shLuc treated cells). Is the hyperphosphorylation of p65 dependent on ILK? How pp65 levels vary upon silencing of any known DENV host-dependency factors (i.e. under reduced viral replication, independently from ILK levels?)

**Part III – Minor Issues: Editorial and Data Presentation Modifications**

Reviewer #1: -line 237 reads "treatment of the ERK inhibitor...", did the authors meant "with ERK inhibitor"?

-please align the blot boxes. This is particularly an issue with Figure 5A.

Reviewer #2: 6) Figure S1 and S7. It may be a bit redundant, but the authors can consider giving S1 its own set of WB images showing that ILK expression is knocked down in the shILK U937 cell line. The knockdown of ILK is also shown in S7 but this is much later in the manuscript.

7) Figure 1: 1A shows there is successful knockdown of ILK. However, authors may consider adding an additional untransduced control cell line for the western blot. The additional western blot could be included in the SI.

8) Figure 1A: The authors should include a column plot that shows the values of AKT and AKT-P473 normalised to actin.

9) Figure 2: Why was Huh-7 used for the replicon experiments? Wouldn’t the A549 be a better cell line compared to the Huh-7 cell line, especially since the Huh-7 cell line may be deficient in some of the interferon pathway components? E.g Huh-7 deficient in MDA5? Or is the A549 cell line non-permissive for the replicon?

10) Figure S2: binding and entry. The authors used an interesting approach to investigate virus binding to the host cell. Instead of quantifying viral RNA using qRT-PCR, the authors performed immunostaining using anti-Env antibodies and analysed the immunostaining using flow cytometry. Can the authors clarify if this method is just as sensitive as the qRT-PCR method? Can the authors also clarify how they defined the gating for S2B? For example, given the gating they used to define Env positive cells, what is the background positive rate from the mock-infected cells? The authors can consider including the mock-infected cells in the graph for S2B.

11) Line 174 to 175: The authors reference some papers and state that “DENV has been known to induce the formation of autophagosomes as replication compartments in cells” Unfortunately, this is an older hypothesis that has been superseded. The DENV replication complex is derived from the ER membrane, not autophagosomes. More recent studies show that induction of autophagy seems to be required for hijacking host lipid metabolism (among other things). The authors should revise this sentence.

12) Figure 6 and for the main text of the manuscript related to figure 6: the authors should elaborate on the HA and flag tagged constructs. Are the HA and flag tags at the N-terminus or the c-terminus? A diagram of the design of the HA and flag tagged constructs might help, especially if the authors can show the location of the ANK repeats, PH domain, and kinase domain in relation to the full length, 1-170, and 171-452 constructs.

13) Figure 7 and discussion related to TNF-a and IL6. Is the decrease in TNF-a and IL6 levels an off target effect of OSU-T315, or is it caused by the lower viral load in the mice?

14) Figure 7D: how were the mouse brain sections prepare after embedding? E.g how were they sectioned? Do the authors have the brightfield images? The nuclei count looks quite different between the sections for the different treatment groups.

Reviewer #3: - Fig5D: the inhibitor (U0126) is used at relatively high concentrations (up to 100uM). Are there any data on the cell viability under these conditions?

-Throughout the manuscript the authors use the wording “ILK enhances DENV infection”, however it might as well act as an host-dependency factor; it would be more appropriate to change the wording..

-Line 163:“ILK fails to affect virus entry”-> ILK does not play a role in virus entry

Fig 2D: please express the % of cell viability as fold-of-control rather than 0.05% TritonX. When displayed as fold of TritonX is difficult to derive whether they were cytotoxic compared to non-treated cells.

PLOS authors have the option to publish the peer review history of their article (what does this mean?). If published, this will include your full peer review and any attached files.

Reviewer #1: No

Reviewer #2: No

Reviewer #3: **Yes: **Pietro Scaturro
---

## [Decision Letter · Decision Letter 1]

30 Nov 2022

Dear Dr. Chang,

Thank you very much for submitting your manuscript "Negative regulation of type I interferon signaling by integrin-linked kinase permits dengue virus replication" for consideration at PLOS Pathogens. As with all papers reviewed by the journal, your manuscript was reviewed by members of the editorial board and by several independent reviewers. In light of the reviews (below this email), we would like to invite the resubmission of a significantly-revised version that takes into account the reviewers' comments.

Thank you for your thoughtful responses and the additional experiments completed to address the reviewers comments. The reviewers still have a few concerns regarding the lack of quantitative experiments for some of the representative experiments that support the main conclusions. If you could please address ideally through experiments but also through re-phrasing conclusions. 

We cannot make any decision about publication until we have seen the revised manuscript and your response to the reviewers' comments. Your revised manuscript is also likely to be sent to reviewers for further evaluation.

Sincerely,

Kellie A. Jurado

Academic Editor

PLOS Pathogens

Stacy Horner

Section Editor

PLOS Pathogens

Kasturi Haldar

Editor-in-Chief

PLOS Pathogens

orcid.org/0000-0001-5065-158X

Michael Malim

Editor-in-Chief

PLOS Pathogens

orcid.org/0000-0002-7699-2064

Thank you for your thoughtful responses and the additional experiments completed to address the reviewers comments. The reviewers still have a few concerns regarding the lack of quantitative experiments for some of the representative experiments that support the main conclusions. If you could please address. Thank you!

Reviewer's Responses to Questions

**Part I - Summary**

Reviewer #1: (No Response)

Reviewer #3: The manuscript is now improved and authors have included some experiments which partially strengthen their conclusions, namely the finding that NS1 and/or NS3 bind to ILK; and that SOCS3 expression is decreased upon ILK silencing. Furthermore, they have now included some data supporting the lack of cytotoxic effects by some of the compounds or shRNA used.

However, this reviewer feels that evidence supporting the modulation of SOCS3 abundance by viral infection by WB or RT-PCR presented (Fig.4A and B respectively; addressed in Point 4 of rebuttal) is still quantitatively inconclusive and would require a quantitative assay (i.e. qRT-PCR; n=3), or a substantial re-phrasing.

**Part II – Major Issues: Key Experiments Required for Acceptance**

Reviewer #1: (No Response)

Reviewer #3: - The additional experiments performed to support the lack of pleiotropic effects underlying SOCS3 reduction presented in Appendix 1 (addressed in Point 5 of the rebuttal) are not conclusive. In this experiment the authors aimed to show that k.d. of an unrelated host protein, would not result in pleiotropic reduction of SOCS3 levels due to decreased viral replication (rather than directly to ILK abundance). However, upon k.d. of ATP6V0C they observe a very mild reduction of DENV replication and only at 72h.p.i. using a quantitative approach (32% reduction in viral replication by qRT-PCR) while conclude that no effects can be observed on SOCS3 expression by WB (performed on lysates at 24.h.p.i.; where no reduction in viral replication can be observed by qRT-PCR). Indeed, by WB one would probably not be able to measure even reduced viral antigen under such mildly reduced viral replication condition. Could the authors show the levels of any viral antigen in this experiment?

- Along these lines, still no further evidence supporting a causal connection between ILK->pp65 is provided (Point 8 of the rebuttal). A variation of 0.6 fold (from 2.0 to 2.6, according to densitometry presented in Fig. 5A; shLuc lanes) in pp65 levels by WB, does not corroborate the notion that ILK is involved in p65 phosphorylation.

Coherently and importantly, the new experiments performed by the authors (Appendix III) clearly show that k.d. of ATP6V0C (a totally unrelated host protein, which also reduces viral replication) reduce levels of pp65 to a very similar levels that ILK k.d.

Altogether, the evidence presented actually consolidate the notion that the reduction of pp65 is unrelated to ILK or SOCS3 levels, but rather a pleiotropic effect of reduced viral replication (conversely the authors conclude that also ATP6V0C – i.e. inhibition of endocytosis - is involved in pp65 regulation).

**Part III – Minor Issues: Editorial and Data Presentation Modifications**

Reviewer #1: (No Response)

Reviewer #3: (No Response)

PLOS authors have the option to publish the peer review history of their article (what does this mean?). If published, this will include your full peer review and any attached files.

Reviewer #1: No

Reviewer #3: No
---

## [Editor Report · Decision Letter 2]

14 Feb 2023

Dear Dr. Chang,

Thank you very much for submitting your manuscript "Negative regulation of type I interferon signaling by integrin-linked kinase permits dengue virus replication" for consideration at PLOS Pathogens. As with all papers reviewed by the journal, your manuscript was reviewed by members of the editorial board and by several independent reviewers. The reviewers appreciated the attention to an important topic. Based on the reviews, we are likely to accept this manuscript for publication, providing that you modify the manuscript according to the review recommendations.

Thank you for addressing many reviewers comments, including the quantitative qPCR probing SOCS3 levels after infection in ILK knockdown cells. Please address why and how the small change (1.4-1.3X) in SOCS3 levels would be the driving factor behind the phenotype (in either text or via experiment)?

Sincerely,

Kellie A. Jurado

Academic Editor

PLOS Pathogens

Stacy Horner

Section Editor

PLOS Pathogens

Kasturi Haldar

Editor-in-Chief

PLOS Pathogens

orcid.org/0000-0001-5065-158X

Michael Malim

Editor-in-Chief

PLOS Pathogens

orcid.org/0000-0002-7699-2064

Thank you for addressing many reviewers comments, including the quantitative qPCR probing SOCS3 levels after infection in ILK knockdown cells. Would you please be willing to explain why the think this small change (1.4-1.3X) in SOCS3 levels would be the driving factor behind the phenotype (in either text or via experiment)?

Reviewer Comments (if any, and for reference):

Figure Files:

Data Requirements:

Reproducibility:

References:

---

## [Editor Report · Decision Letter 3]

25 Feb 2023

Dear Dr. Chang,

We are pleased to inform you that your manuscript 'Negative regulation of type I interferon signaling by integrin-linked kinase permits dengue virus replication' has been provisionally accepted for publication in PLOS Pathogens.

Best regards,

Kellie A. Jurado

Academic Editor

PLOS Pathogens

Stacy Horner

Section Editor

PLOS Pathogens

Kasturi Haldar

Editor-in-Chief

PLOS Pathogens

orcid.org/0000-0001-5065-158X

Michael Malim

Editor-in-Chief

PLOS Pathogens

orcid.org/0000-0002-7699-2064
---

## [Editor Report · Acceptance letter]

10 Mar 2023

Dear Dr. Chang,

We are delighted to inform you that your manuscript, "Negative regulation of type I interferon signaling by integrin-linked kinase permits dengue virus replication," has been formally accepted for publication in PLOS Pathogens.

Best regards,

Kasturi Haldar

Editor-in-Chief

PLOS Pathogens

orcid.org/0000-0001-5065-158X

Michael Malim

Editor-in-Chief

PLOS Pathogens

orcid.org/0000-0002-7699-2064